# Pharmacologic Application Potentials of Sulfated Polysaccharide from Marine Algae

**DOI:** 10.3390/polym11071163

**Published:** 2019-07-08

**Authors:** Joanne Katherine Talens Manlusoc, Chieh-Lun Hsieh, Cheng-Yang Hsieh, Ellen San Nicolas Salac, Ya-Ting Lee, Po-Wei Tsai

**Affiliations:** 1College of Pharmacy, Adamson University, Manila 1000, Philippines; 2Department of Athletics Sports, College of Humanities and Social Sciences, Chang Jung Christian University, Tainan 711, Taiwan; 3Graduate Institute of Medical Sciences, College of Health Sciences, Chang Jung Christian University, Tainan 711, Taiwan; 4Office of the Vice President for Academic Affairs, Emilio Aguinaldo College, Manila 1000, Philippines; 5Department of Beauty Science, National Taichung University of Science and Technology, Taichung 404, Taiwan; 6Department of Medical Sciences Industry, College of Health Sciences, Chang Jung Christian University, Tainan 711, Taiwan

**Keywords:** sulfated polysaccharides, marine algae, seaweeds, bioactivity, pharmacologic activity, therapeutics, natural polymers

## Abstract

With the advent of exploration in finding new sources for treating different diseases, one possible natural source is from marine algae. Having an array of potential benefits, researchers are interested in the components which comprise one of these activities. This can lead to the isolation of active compounds with biological activities, such as antioxidation of free radicals, anti-inflammation, antiproliferation of cancer cells, and anticoagulant to name a few. One of the compounds that are isolated from marine algae are sulfated polysaccharides (SPs). SPs are complex heterogenous natural polymers with an abundance found in different species of marine algae. Marine algae are known to be one of the most important sources of SPs, and depending on the species, its chemical structure varies. This variety has important physical and chemical components and functions which has gained the attention of researchers as this contributes to the many facets of its pharmacologic activity. In this review, recent pharmacologic application potentials and updates on the use of SPs from marine algae are discussed.

## 1. Introduction

Over the past decade, studies about the promising pharmacologic potential and importance of marine algae or seaweeds are increasing. Marine algae attributes most of its biological activity to a natural polymer that can be isolated from it which is known as sulfated polysaccharides (SPs) [1]. SPs are complex heterogenous natural polymers with an abundance found in different species of marine algae. Depending on whether it is isolated from green (Chlorophyta), brown (Phaeophyta) or red (Rhodophyta) algae, its chemical composition and structure varies [2]. These variabilities, including the presence and quantitative contents of carbohydrates, proteins, sulfate and its degree of sulfation, can yield different pharmacologic applications that have been tested in vitro, ex vivo and in vivo in animal models with promising results. These results may be of benefit in treating different diseases and can be possibly used as antioxidant, anti-allergy, antiviral, anticancer, and anticoagulating agents [3]. This review discusses recent updates on SPs extracted and isolated from marine algae and its increasing potential as a pharmacological agent. 

## 2. Pharmacologic Potential Activities of SPs

### 2.1. Anti-Inflammatory Activity

In a study of Sanjeew et al [4], high molecular weight SPs were isolated from *Sargassum horneri* and tested both in vitro in RAW 264.7 cells and in an vivo model in zebrafish. The highest molecular weight fraction yielded inhibitory activity on lipopolysaccharides (LPS)-induced nitric oxide (NO) in RAW 264.7 macrophage cells. It was also observed that prostaglandin E2 (PGE_2_) production and pro-inflammatory cytokine production was also inhibited on a proposed mechanism of downregulation of the KB signaling cascade. LPS-induced zebrafish also showed NO inhibitory activity in the same fraction (IC_50_ = 87.12 μg/mL). An earlier research performed on the same brown algae concluded that one of the purified fractions caused a significant decrease in the tumor necrosis factor (TNF-α) secretion. Further, interleukin-10 (IL-10) secretions were seen in LPS-induced RAW 264.7 cells and [5] NO release and inducible NO synthase (iNOS) were inhibited by the SP fraction at a high dose at 200 μg/mL. To further understand the possible mechanism of the action of fucoidan on its elicited inflammatory response, fucoidan from brown algae *Saccharina latissimi* and *Fucus versiculosus* [6] were tested on polymorphonuclear neutrophils (PMN) and their ability to bind to the target proteins closely linked to the anti-inflammatory process. Both fucoidans isolated in the algae showed affinity on both IL-8 and C5a with a higher affinity on the former. Another observed reduction was also seen on the IL-8 and C5a-induced intracellular calcium release, extracellular signal-regulated kinase (ERK) 1 and 2 phosphorylation, and chemotaxis of PMN which showed that it was not the only target protein responsible for the response. Fucoidans from *Choonospora minima* [7] showed a very similar activity on a zebrafish model and RAW 264.7 macrophages cells as they inhibited anti-inflammatory factors, such as NO and reactive oxygen species (ROS) production (IC_50_ = 27.82 ± 0.88 μg/mL), PGE_2_ expressions, TNF-α, IL-1β and IL-6. *S. cristaefolium* [8] also showed similar activity on the same cell model with regards to the inhibition of iNOS expression by inhibiting phosphorylation of the signal protein in the model. An in vivo model was used to test the SPs isolated from *S. hemiphyllum,* however, the SPs still elicited a similar response in relation to the effect of anti-inflammation on in vitro models for the doses of SPs at 40 mg/kg and 80 mg/kg [9]. The SPs were able to reduce the ear swelling and erythema in two animal models by means of decreased pro-inflammatory factors, such as NO, myeloperoxidase, IL-1β, IL-6, and TNF-α. These studies solidified the conception that inhibition and downregulation of pro-inflammatory factors has been the key in suppression of inflammation as one of the mechanistic approaches of SPs. Table 1 below summarizes the recent studies mentioned and other algal sources with SPs anti-inflammatory studies.

### 2.2. Antiangiogenic Activity

Chick chorioallontoic membrane (CAM) assay, an in ovo method, is a technique used to observe angiogenesis. The fractions extracted from several Phaeophyceae species have exhibited antiangiogenic properties in the CAM assay. In the study of Kurup et al [19], the purified sulfated polysaccharides were isolated from *Padina tetrastromatica*, then named as ESPs-CP (IC_50_ = 1.2 mg/mL) and showed a significant suppression of angiogenesis, and a reduction in hemoglobin was also quantified. This was further confirmed by the downregulation of an angiogenic mediator, VEGF. HIF1A expression in HeLa cells was also performed in the study. On a same model, fucoidans from *Turbinaria conoides* showed a dose-dependent inhibition on the formation of blood vessels in a chick embryo. The reduction in the total number of tubules and its length and sizes were also observed as well as total number of tubular junctions [20]. Both fractionated and purified anionic SPs from *Sargassum vulgare* showed strong antiangiogenic properties as compared to the standard heparin. This result showed that PSV1 at 50 μg/μL (71.4%) was more effective than SV1 at 100 μg/μL (75.9%). It was also supported by the an inhibition of tubulogenesis in rabbit aorta endothelial cells [21]. These results are confirmed by the antiangiogenic properties of SPs from *Laminaria japonica* [22] and *Sargassum fusiforme* [23] on the in vitro angiogenesis model where formation of tubules were inhibited in both species. Angiogenesis plays an important role in tumor growth, metastasis and inhibiting vascular growth as these abnormalities can lead to a positive impact in people with tumors. These findings suggest that the antiangiogenic activity of SPs holds a promising novel approach in trying to develop anticancer drugs, and further, they are good indicators that marine algae sourced SPs can be of beneficial use. Table 2 below summarizes the mentioned studies and other recent studies about SPs antiangiogenic activity.

### 2.3. Antibacterial Activity

In the recent years, the use of SPs in synthesizing nanoplatforms has been explored. The extracted and isolated SPs were mostly used as reducing and stabilizing agents to yield inorganic metallic nano-platforms as possible drug delivery systems to elicit their antibacterial activities. Aqueous extracted SP from *Sargassum wightii* [25] was used to synthesize magnesium oxide nanoparticles (MgONPs). The synthesized MgONPs exhibited bactericidal activity on gram-negative bacteria, specifically the highest in MRSA 56 and gram-positive bacteria with the highest inhibitory activity in *Pseudomonas aeruginosa.* Ulvans found in *Ulva armoricana* [26] that were used to synthesize silver nanoparticles (AgNP) were tested in bacterial strains of *Staphylococcus aureus, S. epidermidis, Escherichia coli,* and *P. aeruginosa*. AgNP showed IC_50_ = 10 μg/mL in Balb/3T3 mouse embryo fibroblasts and antimicrobial activity. Further rapid bactericidal activity was observed on gram-negative bacteria, however, all tests showed that the synthesized AgNP with ulvan exhibited stronger antibacterial property compared to the free ulvan. The same effects were also observed in the algae *Ulva rigida* in which the nanocomplex carrier formed with the isolated ulvan exhibited higher inhibitory activity in *S. aureus* compared to the free ulvan. The dose of ulvan at 20 μg/mL of *S. aureus* showed bacterial growth inhibition (9%) treated with lysozyme (200 μg/mL) [27]. The property of the nanocarriers that is highly attributed to its antibacterial activity is its nanosized dimension which enhances the permeability through the membranes of the bacterial strains. Table 3 below summarizes the mentioned studies and other recent studies about SPs being utilized directly as an antibacterial agent.

### 2.4. Anticoagulant Activity

Green algae, in the past few years, have been strongly linked to its anticoagulant activity. A novel rhamnan-type SP was isolated in the green algae *Monostroma angicava* [32,33]. The fractions of the SP were shown to have good responses to its anticoagulant parameters such as activated partial thromboplastin time (APTT) and thrombin time (TT). However, the fraction showed a weak effect on factor Xa inhibition. This suggests that only strong potentiation thrombin by heparin cofactor II occurs. In rat models, high fibrin(ogen)olytic activity was seen. Common on the following green algae *M. oxysperma* [34], *Penicillus captatis* [35], *Codium divaricatum* [36], and *Enteromorpha linza* [37], APTT were prolonged signifying an inhibition in the intrinsic and/or common pathway of coagulation. This means that it lowers the risk of bleeding. All but *M. oxsperma* was affirmative of a prolonged TT. Prolonged TT inhibits thrombin activity. Prothrombin time (PT) was also devoid on the above mentioned green algae. All four studies suggest that the sulfate content present in the SP is responsible for similar anticoagulant activities that green algae possess. Table 4 below summarizes the mentioned and additional studies about SPs anticoagulant activity.

### 2.5. Antidiabetic Activity

For years, SPs have been considered as a functional food due to their promising antidiabetic properties. Several brown algae from marine sources have been studied to be effective on alloxan-induced diabetic rodents as an antidiabetic. SPs isolated from marine algae *Sargassum vulgare*, showed a remarkable correction on the fasting and postprandial blood glucose of diabetic rats after administering the SP intraperitoneally for 30 days. The SVSP administration indicated a significant decrease in concentrations of serum for T-Ch, TG, LDL-Ch (24.91%, 46.78% and 51.75%, respectively) in the animal model for diabetics. This is a significant increase related to the serum HDL-Chconcentration (53.12%) compared to untreated diabetics [44]. This study showed an inhibition of pancreatic a-amylase, an enzyme responsible for the increase of blood sugar levels. Other beneficial effects of the SP isolated in diabetic rats includes a decrease in HBA1c, improved hemogram parameters, such as white blood cells, platelets, lymphocytes, and parameters that lower in diabetic rats that can possibly cause anemia. Apart from these factors, it was also observed on the histopathological findings that the liver and kidneys of these SP treated diabetic rats were protected. *Saccharina japonica* isolated fucoidan has also seen a reduction of blood glucose levels of alloxan-induced rats in some concentrations of the treatment. Insulin levels were also known to increase, and serum lipid levels improved. An administration of fucoidan at 200 and 1200 mg/kg body weight/day could significantly reduce blood glucose levels by 22% and 34%, respectively. The oral glucose tolerance test (OGTT) showed some effect on glucose disposal in trials after 15 days of SP treatment. This was an improvement in glucose tolerance in the treatment group. These effects led to a possible increase in insulin sensitivity or reduction in insulin metabolism [45]. A-glucosidase, an enzyme responsible for the breakdown of complex sugars that leads to an increase in blood sugar, is also one of the targets of some antidiabetic treatments. SP from *Ascophyllum nodosum* [46] and *Sargassum pallidum* [47] have shown a-glucosidase inhibition in some concentrations of its sulfate derivatives. *S. pallidum* at the high concentrations of 1 mg/mL showed the highest inhibitory rates of SPP, S-SPP_1–4_, S-SPP_1–6_ and S-SPP_1–8_ which were 76.3%, 96.9%, 97.6% and 98.4%, respectively. The inhibition of sulfated derivatives showed to be much higher than native SPP. On *A. nodosum* SP-treated cell models, glucagon-like peptide-1 (GLP-1) showed an increased secretion. GLP-1 promotes glucose-dependent insulin secretion from the pancreatic b-cells. Human OGTT also revealed in 8-week ingestion that the SP caused lower glucose in the treatment group compared to the placebo group. On both of these studies on marine algae, the researchers attributed it to the sulfate derivative of the polysaccharide, but further structure-activity relationship studies are needed to prove this. Table 5 below summarizes the mentioned and recent studies about SPs antidiabetic potential.

### 2.6. Antioxidant Activity

It is well known that free radicals include oxygen free radicals and non-radical derivatives of oxygen. They are the byproducts of normal metabolism. Superoxide and OH-radicals are the important reactive oxygen species in the body. One of the main causes of cytotoxicity is superoxide free radicals, because they are the first oxygen free radicals produced in the body and last longer than other free radicals. Hydroxyl groups are the most active free radicals that attack all biomolecules by initiating a free radical chain reaction. However, excess free radicals are potentially harmful to various biomolecules through lipid peroxidation, DNA damage and inhibition of protein synthesis. This damage accelerates aging and causes many diseases, such as cancer and tumors [50]. Many genera of algae have been reported regarding their antioxidant activity. The algae antioxidants which include phycobiliproteins, phlorotannins, carotenoids, sulfated polysaccharides, scytonemin and mycosporin-like amino acids also have been shown in the literature by their biologically significant activities. Fucoidan, porphyran, carrageenans, and ulvan (Figure 1) are part of SPs which have been found for their antioxidant activity, such as DPPH, ABTS, NO, super oxide and hydroxyl radicals.

One of the most attributed properties of marine algae SP is its antioxidant potential. Many studies have been undertaken to prove this property through different assays. Superoxide radical assay, lipid peroxidation inhibition, hydroxyl radical assay, DPPH radical scavenging assay, total antioxidant assay, ABTS radical scavenging activity, reducing power assay, ferrous ion chelating ability, ferric ion reduction, photochemiluminiscence (PCL)-radical scavenging, and H_2_O_2_ oxidative stress inhibition on cells are the most common assays to measure this potential. These assays test the viability of SPs to inhibit oxidation than can form free radicals and ROS that damages cells that can lead to negative possible outcomes in health [51]. Only natural products (compounds, extracts) have antioxidant properties. This is a good indicator that these marine algae species have a potential to be turned into viable pharmacologic agents as well as additives in different products ranging from food to cosmetic products. Table 6 below summarizes recent studies about SPs antioxidant activity.

### 2.7. Antiviral Activity

In the study of Thuy et al. [84], three species of brown algae isolated with fucoidan all possessed anti-HIV activity in the cell lines. *Sargassum mcclurei, S. polycystum,* and *Turbinaria ornate* showed similar activities and showed no cytotoxicity on normal cells. In the study, the sulfate groups presented in the backbone of the fucoidan structure was not responsible in the activity unlike other SP activities. They attributed the possible mechanism of action on the inhibition by the HIV-1 infection entry on the host cell and not during the infection, pointing out the blockade on the early step HIV entry. The oral administration of fucoidan from *Undaria pinnatifida* in immunocompetent and immunocompromised mice and in vitro showed a reduced viral replication of influenza A virus. The IC_50_ range was 27–170 and 90–120 mg/mL in the isolates and challenge virus, respectively. The subpopulation presence of drug-resistant viruses was confirmed in the lungs and bronchoalveolar lavage fluids of immunocompromised mice administered with oseltamivir. The IC_50_ range was 0.16–42 and 0.18–0.30 mg/mL in the isolates and challenge virus, respectively [85]. This activity led to mortality and stimulated immune defenses of the virus-infected mice. The fucoidan was also co-administered with a neuraminidase inhibitor oseltamivir. The advantage of the fucoidan over oseltamivir is an increase in the neutralizing antibody found in the mucosa and blood of the mice. It is also noted that there is an absence of recovered resistant virus in the treatment of oseltamivir alone, while none was fionid on the combination. This exhibits a promising candidate for novel therapeutics that can possibly lessen the resistance to retroviruses. Table 7 below summarizes the mentioned and recent studies about SPs antiviral activity.

### 2.8. Anticancer Activity

Several cell lines have been tested on the potential of SPs from marine algae and holds association to the possibility of these natural polymers to be a radical novel treatment for cancer. Two species of brown algae were tested for its apoptotic activity with HeLa cervical cancer cell line. *Laminaria japonica* crude fucoidan and fractions exhibited some apoptotic activity exhibited in the number of viable cells. Regarding CFs concentration and their fractions, it was found that the cells could be dead. An exhibition of F3 showed less cytotoxicity than F1 and F2 with approximately 30% inhibition at the concentration of 2 mg/mL. The literature also indicated that F1 and F2 of the extract fucoidan isolated from *Laminaria japonica* has been found to have less activity than the fucoidans from *Undaria pinnatifida* with the IC_50_ = 0.4 mg/mL [69]. They associated the activity in the presence of sulfate and fucose, thus leaving the fraction with a few sulfates to be less apoptotic than the other fractions. The researchers ruled out that this attribute was associated with the kind of marine algae species being utilized. *Padina tetrastromatica* has also been extensively tested on HeLa cells on genes that have most likely been upregulated and downregulated to induce cancer cell death. The ESPs-CP decreased the HeLa cells viability in a dose-dependent manner and at 1.2 mg/ml concentration showed a 50% reduction (IC_50_) in the viability [19,88]. A reduction of viable HeLa cells occurred at 50–55% when treated with the SP. Treated cells also showed ROS generation and mitochondrial membrane depolarization in almost half of the cancer cells which suggests that one of the mechanistic approaches on the apoptosis of the cell is by a ROS-mediated intrinsic pathway. This is seen on the upregulation of pro-apoptotic mRNA gene expression and downregulation of antiapoptotic genes. An increase in the expression of mediators of apoptosis was also present. SPs from *P. tetrastromatica* have also successfully arrested cell cycles at the G0/G1 phase apparent by the upregulation of P21 gene which is a major G1 phase regulator. The downregulation of CCDN1 and AURKB were also seen. Cancer cell migration was also non-apparent as it was seen that it did not occur in treated cells in a scratch test and downregulation of MMP-2 and MMP-9 genes which mediates metastasis.

In A549 lung cancer cells, MgONPs synthesized with a brown algae *Sargassum wightii* showed a dose-dependent cytotoxicity in a MTT assay, although a possible mechanism was not proposed for its cytotoxicity. MgONPs showed cytotoxicity against A549 cell line in a dose dependent manner with the IC_50_ = 37.5 ± 0.34 μg/mL [25]. Red algae *Acanthopora specifica* purified fraction has been tested in the cell line and the exhibition of cytotoxicity at the range of concentrations at 100 to 1000 μg/mL [52] were found. The cytotoxicity was attributed to the destruction of the monolayer membrane of the A549 cancer cells which were visible morphologically and apoptosis fluorometrically. *S. palgiophyllum* SP fractions were also cytotoxic in MTT assays attributing apoptotic cell death by cell cycle arrest at the G0/G1, G2/M, and S phases. The F2 fraction showed more effective anticancer activity in both of HepG2 and A549 cells with IC_50_ values of 600 μg/mL and 700 μg/mL, respectively [89]. On AO/EtBr staining, morphologically it exhibited both apoptotic and necrotic bodies. This proves a cell cycle-regulating mechanism of the SP on A549. This attributes the properties variability on the sulfate content of the SP and other structural parameters that differ in each isolated fraction.

Apparently, *S. palgiophullum* SP exhibited a similar activity with the hepatocellular cancer cell line HepG2 [89]. The fractions of HE1, HE4 and HE7 were obtained from polysaccharide of EHEM and showed the cytotoxicity against HepG2 with IC_50_ = 73.1, 42.6 and 76.2 μg/mL, respectively. A dose-dependent cytotoxicity by apoptotic and necrotic mechanism was seen. Other marine algae *Jania rubens, Pterocladia capillaceae,* and *Enteromorpha intestinalis* [24] also were tested for HepG2 cell line cytotoxicity. While purified fractions of *E. intestinalis* was also seen as a viable cell line cytotoxic agent, purified fractions of *J. rubens* and *P. capillaceae* showed no to weak cytotoxicity. This was possibly a synergistic effect as the effect in a hepatocellular carcinoma rat model showed a significant decrease of a-fetoprotein, carcinoembryonic antigen, glypican-3, and hepatocyte growth factors which were elevated in hepatocarcinogenesis.

DLD-1 colon cancer cell line growth was seen to be inhibited by a fraction isolated from *S. horneri*. The polysaccharide fractions of SHP30 and SHP80 showed some inhibition effects at the concentrations from 0.5 to 6.0 mg/mL. The SHP30 fraction content had the highest sulfate and intermediate molecular weight which showed relatively higher inhibition of 51.92% at the concentration of 2 mg/mL than SHP80 at all concentrations. Otherwise, the SHP60 fraction conduced MKN45 cells to grow at concentration ranges from 0.5 to 6 mg/mL [90]. The fraction with the highest sulfate content has caused the highest inhibitory effect on the cancer cell lines. The cytotoxicity was also attributed to apoptosis cell death at the G0/G1 and S phase of the cell cycle which is also suggestive of the upregulation of Bax and downregulation of Bcl-2 gene in the fraction treated SP group. These data show that, not only does sulfate content play an important role in eliciting its anticancer activity, but also the way it targets proteins in the cancer cell. Another brown algae Fucoidan from *Saccharina cichorioides* have exhibited inhibition in the proliferation of the cell lines as well. Table 8 below summarizes the mentioned and other recent studies about SPs anticancer studies on different available cancer cell lines and tumors.

### 2.9. Gastroprotective Activity

Isolated SPs from 3 species of red algae have been attributed to work as a gastroprotective agent. The SP in *Laurencia obtuse* possess a high molecular weight sulfated polysaccharide. The gastroprotective effect (*p* < 0.01) of SPs has been also observed with the gastric ulcer inhibition of 63.44%, 78.42% and 82.15% at concentration ranges of 25, 50 and 100 mg/kg, respectively [71]. It reduced gastric ulceration in a dose-dependent manner giving a significant gastroprotective effect. In ethanol/HCl-induced gastric mucosa in rats, the SP showed significant increases in glutathione (GSH) levels in stomach tissues and decreased levels of thiobarbituric acid-reactive substances that are usually seen as a by-product of lipid peroxidation. This is an indication that the mechanism of action of the possible gastroprotection is through a decrease in oxidative stress. The following is true for the high molecular weight SPs isolated from *Hypnea musciformis*. An in vivo model was treated with sulfated-polysaccharide (PLS) fractions of *H. musciformis* at the doses of 3, 10, 30, and 90 mg/kg. Distal colon excised from a trinitrobenzene sulfonic acid-induced intestinal damaged rats have seen a reduction in colitis and levels of biochemical parameters, such as GSH which showed an increase and malonyldialdehyde acid (MDA) showed a decrease [14]. Further, ethanol-induced gastric damage in mice have also seen an increase on GSH and a decrease in MDA which has reversed gastric injury upon administration of SP in a dose-dependent matter attributing this to the possible activation of NO/KATP pathway [94], another oxidative stress reduction mechanism. On the same ethanol-induced gastric damage model in rodents, SP with a high degree of sulfation from *Solieria filiformis* possess a very similar mechanism, still attributing it to the decrease in the production of free radicals and a decrease in hemoglobin concentration in the gastric mucosa being an indirect marker of its gastroprotective property [79]. Table 9 below summarizes the mentioned studies about SPs gastroprotective activity.

### 2.10. Immunomodulatory Activity

The activation of macrophages is one of the indicative factors that immunomodulatory activity takes place and is one of the biggest pharmacologic contributions of SPs. SPs found in different species of the family Phaeophyta were studied for their effects on RAW 264.7 macrophage cells. Their immunomodulatory properties can be further studied to enhance defenses in pathogenic and invasive cells in the body. *Cystoseira indica* SP fractions with the highest sulfate content showed enhanced phagocytic activity of the macrophage cell. Acid phosphatase activity and NO production significantly increased. This leads to an imperative that the SP fraction from *C. indica* is an immunostimulant by means of cell activation and cytokine secretion. All crude and fractionated polysaccharides of *C. indica* were tested on RAW 264.7 cells at concentration ranges from 10 to 50 μg/mL. CIF2 polysaccharide is a proliferation stimuli on RAW 264.7 cells, and can enhance cell growth to approximately 25% [94]. Another brown algae *Dictyopteris divaricata* showed a similar immunostimulant activity. The proliferation of RAW 264.7 macrophages were observed and production of NO significantly increased. The DDSP were a proliferation stimuli on RAW 264.7 cells at the concentration ranges of 50 to 400 μg/mL [59]. *Padina tetrastromatica* SP, with a similar activity on the macrophage, showed proliferation of the macrophage cells. Inflammatory mediators, 5-LOX, COX-2, and PGE_2,_ have also shown an increase as well as NO production and iNOS activity in a dose-dependent manner comparable with LPS implying an immunostimulant effect again of the SP [87]. *Sargassum angustifolium* SP crude and fractions were all tested to have an immunostimulant effect on the RAW 264.7 cells as they profilerated the macrophage and at the same time increased NO production, having one of the fractions with upregulation in the iNOS mRNA expression. The crude and fractions of *S. angustifolium* had effects on the proliferation of RAW 264.7 cells at concentrations ranges from 10 to 50 μg/mL. Fraction F2 was a proliferation stimuli on RAW 264.7 cells at concentrations of 50 μg/mL (p < 0.05) [96]. Proinflammatory cytokines were also shown to increase with this trend. To further assess the mechanism of the effects, the most active fraction showed an increase in the levels of NF-kB in the cytoplasm. These effects were attributed to the NF-kB signaling pathway in activating the macrophage cell. Ascophyllan SP showed a different immunostimulant pathway in the macrophage cells. ROS generation was measured and was shown to have an increase in the ascophyllan-induced cells. To further confirm the immunostimulant effect, levels of cytosolic subunits were shown to increase in the plasma membrane. The ROS production was confirmed to be elicited by the JNK MAP kinase signaling pathway as levels of it increasing were observed in incubated cells with treated ascophyllan. The phosphory-lated ERK expression showed no significant differences at the ascophyllan concentration ranges of 0 to 1000 μg/mL [50]. Notably, SP from *Ecklonia cava* has seen a similar response by activating lymphocytic activation with the JNK/NFkB pathway similar to the previously mentioned mechanisms. Some studies have indicated that SP led to the lymphocytes proliferation at the concentration ranges of 25, 50, and 150 μg/mL [97]. Table 10 below summarizes the mentioned and recent studies about SPs immunomodulatory activities.

### 2.11. Other Pharmacologic Activities

Other pharmacologic activities attributed to the presence of SPs in marine algae have been documented, especially in vivo rodent models. The efficacy of SPs from *Hypnea musciformis* (PLS 90 mg/Kg) [108] and *Gracilaria cervicornis* (PLS of enzymatic extraction 25 to 30 g) [109] as antidiarrheal were tested in rats and mice respectively. In both studies, pre-treatment of SP fractions on castor oil-induced and cholera toxin-induced rodents showed fecal and diarrheal stool decreases in output, an increase in Na + /K+ ATPase activity in the small intestine, and a decrease in both the cholera toxin-induced fluid secretion and small intestine chloride excretion. This may be attributed to the reduction of the toxin possibly binding to the GM1 receptor. However, SPs isolated from *G. cervicornis* did not show any effect on the gastrointestinal motility compared to *H. musciformis,* which may attribute its activity on the cholinergic receptor activation to the mechanism of action, similar to the drug, loperamide.

Recent studies also suggest that *Enteromorpha prolifera* (the administration of EP 200 mg/kg body weight/daily) [110] and *Cystoseira crinita* (the administration of CCSP 200 mg/kg of body weight/daily) [111] both possess antihyperlipidemic in rats. High fat diet-induced hyperlipidemic rats were administered with SPs and both marine algae significantly reduced the weight of the rats. SPs from *E. prolifera* reduced the liver index and serum triglyceride levels. This can possibly be attributed to the mechanism of action on the decreased mRNA, suppressed SREBP-1 and ACC expressions which have been responsible for fatty acid synthesis. The reduction on the levels of triglycerides can also be attributed to a reduced risk of cardiovascular disease, indicating that it can be a good cardioprotective. *C. crinita* were shown to decrease in low density lipoproteins and triglycerides, and increase in high density lipoproteins in the blood levels of the rats. Aside from its antihyperlipidemic activity, *C. crinita* also exhibited possible hepatoprotective and renoprotective activities. This is due to its liver-kidney function efficiency and positive effects on the histological analysis of the rats administered with SPs.

*Turbinaria decurrens*, a brown alga, has been attributed to possess hepatoprotective and neuroprotective activities in both rat and mice studies respectively [112,113]. Fucoidan isolated from *T. decurrens* was tested on ethanol-induced hepatotoxic rats and was shown to have hepatoprotective properties due to its antioxidative properties in which oxidative stress was seen to be high in ethanol-induced rats. The fucoidan also decreased levels of lipid peroxidation markers. Fucoidan found in *T. decurrens* have been seen to have some neuroprotective activities on 1-methyl-4-phenyl-1,2,3,6-tetrahydropyridine (MPTP)-induced mice, a known prodrug to a neurotoxin that causes Parkinson’s disease. In the study, the treatment group with fucoidan showed an increase in dopamine levels compared to the group without the SP. The fucoidan also showed a reduction and reversed expression of tyrosine hydroxylase and dopamine transporters in MPTP-induced groups, which shows its neuroprotective activity.

Other marine algae, such as *Sargassum vulgare*, showed antithrombosis and anti-inflammatory responses in rat models. The SV1 of *S. vulgare* was evaluated for inflammatory activity at the concentration ranges of 10, 30 and 50 mg/kg using a carrageenan-induced acute rat model [43]. *Laminaria japonica* was also evaluated as a renoprotective in glycerol-induced acute injury rats by administering intraperitoneally low molecular weight -sulfated polysaccharides [114] and was seen to be a vascular protective on adrenaline-induced vascular endothelial damage in rats after being subjecting to physiological stress [115]. *Sargassum fusiforme* lessened UVB-induced oxidative stress on Kun Ming mice, and has promising use as a cosmetic skin protective [116]. Table 11 below shows a summary of other pharmacological activities attributed to SPs in different marine algae sources. This also includes studies done in ex vivo and in vitro models.

## 3. Conclusions

In this review, the different therapeutic possibilities of SPs from marine macroalgae in different in vitro, in vivo, in ovo, and ex vivo models via different techniques and methods was discussed. The differences in chemical composition, especially the sulfate content and versatile linkages, brought about by the myriad of methods in isolating and forming structures in the many species of marine algae, still did not produce a one-size-fits-all mechanism of this natural polymer. There are studies for researchers to attribute one activity of the SPs on a defined SP makeup. The trend continues to understand a molecular mechanistic approach on how these natural polymers work. While variabilities in these compositions still pose a hindrance in understanding its pharmacologic attributes, these variabilities also give endless possibilities on how to address problems in several diseases with different mechanisms of action.

## Figures and Tables

**Figure 1 polymers-11-01163-f001:**
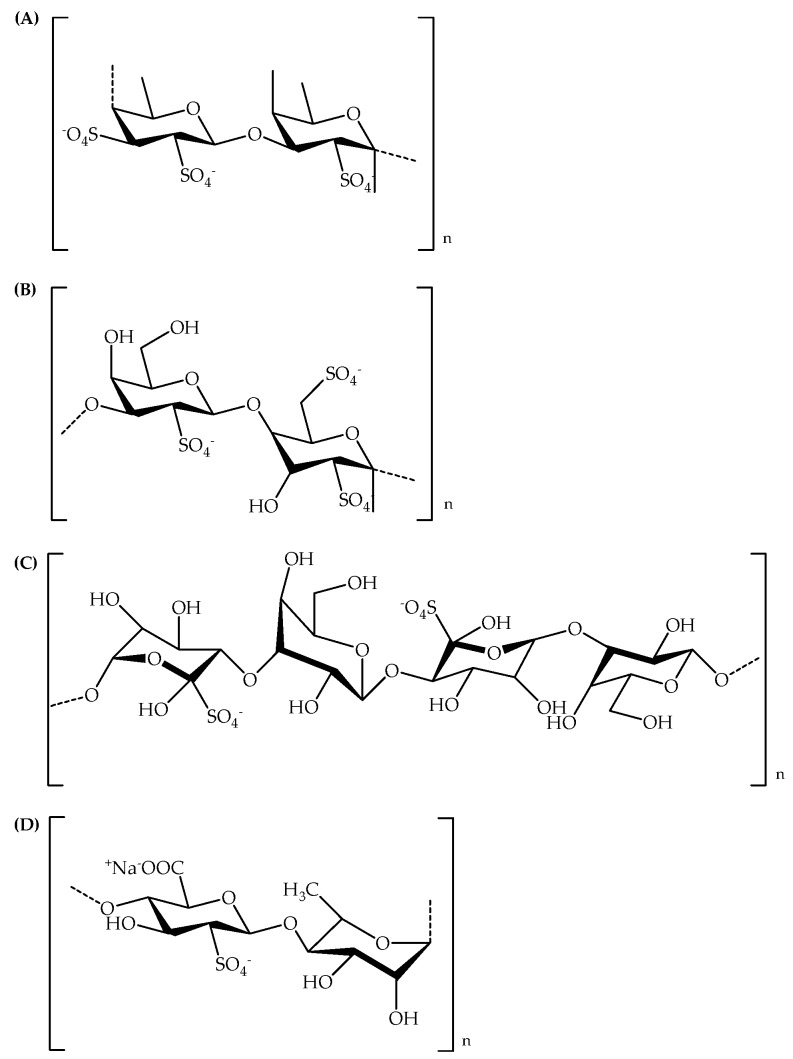
Sulfated polysaccharide chemical structures of (**A**) Fucoidan, (**B**) Carrageenan, (**C**) Porphyran, (**D**) Ulvan.

**Table 1 polymers-11-01163-t001:** Marine algae source with anti-inflammatory activity attributed to sulfated polysaccharides (SPs).

Marine Algal Source	Algae Group	Reference
*Achanthophora muscoides*	Red algae	[10]
*Agardhiella ramosissima*	Red algae	[11]
*Ahnfeltiopsis flabelliformis*	Red algae	[12]
*Choonospora minima*	Brown algae	[7]
*Fucus vesiculosus*	Brown algae	[6]
*Gracilaria cornea*	Red algae	[13]
*Hypnea musciformis*	Red algae	[14]
*Padina australis*	Brown algae	[15]
*Porphyra yezoensis*	Red algae	[16]
*Sargassum binderi*	Brown algae	[15]
*Sargassum cristaefolium*	Brown algae	[8]
*Sargassum duplicatum*	Brown algae	[15]
*Sargassum fulvellum*	Brown algae	[15]
*Sargassum hemiphyllum*	Brown algae	[9]
*Sargassum hornei*	Brown algae	[4,5]
*Saccharina latissima*	Brown algae	[6]
*Solieria filiformis*	Red algae	[17]
*Turbinaria turbinata*	Brown alagae	[15]
*Ulva lactula*	Green algae	[18]

**Table 2 polymers-11-01163-t002:** Marine algae source with antiangiogenic activity attributed to SPs.

Marine Algal Source	Algae Group	Reference
*Enteromorpha intestinalis*	Green algae	[24]
*Jania rubens*	Red algae	[24]
*Laminaria japonica*	Brown algae	[22]
*Padina tetrastromatica*	Brown algae	[19]
*Pterocladia capillacea*	Red algae	[24]
*Turbinaria conoides*	Brown algae	[20]
*Sargassum fusiforme*	Brown algae	[23]
*Sargassum vulgare*	Brown algae	[21]

**Table 3 polymers-11-01163-t003:** Marine algae source with antibacterial activity attributed to SPs.

Marine Algal Source	Algae Group	Reference
*Caulerpa racemosa*	Green algae	[28]
*Gracilaria corticata*	Red algae	[29]
*Laminaria japonica*	Brown algae	[30]
*Sargassum wightii*	Brown algae	[25]
*Spatoglossum asperum*	Brown algae	[31]
*Ulva armoricana*	Green algae	[26]
*Ulva rigida*	Green algae	[27]

**Table 4 polymers-11-01163-t004:** Marine algae source with anticoagulant activity attributed to SPs.

Marine Algal Source	Algae Group	Reference
*Ahnfeltiopsis flabelliformis*	Red algae	[38]
*Codium divaricatum*	Green algae	[36]
*Enteromorpha linza*	Green algae	[37]
*Gracilaria debilis*	Red algae	[39]
*Grateloupia livida*	Red algae	[40]
*Mastocarpus stellatus*	Red algae	[41]
*Monostroma angicava*	Green algae	[32,33]
*Monostroma oxysperma*	Green algae	[34]
*Penicillus capitatus*	Green algae	[35]
*Sargassum aquifolium*	Brown algae	[42]
*Sargassum vulgare*	Brown algae	[43]

**Table 5 polymers-11-01163-t005:** Marine algae source with antidiabetic activity attributed to SPs.

Marine Algal Source	Algae Group	Reference
*Ascophyllum nodosum*	Brown algae	[46]
*Codium fragile*	Green algae	[48]
*Laminaria japonica*	Brown algae	[49]
*Monostroma angicava*	Green algae	[33]
*Saccharina japonica*	Brown algae	[45]
*Sargassum pallidum*	Brown algae	[47]
*Sargassum vulgarae*	Brown algae	[44]

**Table 6 polymers-11-01163-t006:** Marine algae source with antioxidant activity attributed to SPs.

Marine Algal Source	Algae Group	Reference
*Acanthophora spicifera*	Red algae	[52]
*Ascophyllum nodosum*	Brown algae	[53]
*Bryopsis* *plumose*	Green algae	[54]
*Caulerpa cupressoides* var. *flabellata*	Green algae	[55]
*Chlorophyta sertularioides*	Green algae	[56]
*Cystoseria barbata*	Brown algae	[57]
*Cystoseira compressa*	Brown algae	[58]
*Dictyota cervicornis*	Brown algae	[56]
*Dictyopteris delicatula*	Brown algae	[56]
*Dictyopteris divaricata*	Brown algae	[59]
*Enteromorpha linza*	Green algae	[54,60]
*Enteromorpha prolifera*	Green algae	[61]
*Fucus vesiculosus*	Green algae	[62]
*Gloiopeltis furcata*	Red algae	[63]
*Gracilaria caudata*	Red algae	[64]
*Gracilaria corticata*	Red algae	[29]
*Gracilaria debilis*	Red algae	[39]
*Gracilaria rubra*	Red algae	[65]
*Gracilariopsis lemaneiformis*	Red algae	[66]
*Grateloupia livida*	Red algae	[40]
*Hizikia fusiforme*	Brown algae	[67]
*Laminaria japonica*	Brown algae	[54,68,69,70]
*Laurencia obtusa*	Red algae	[71]
*Mastocarpus stellatus*	Red algae	[41]
*Monostroma oxyspermum*	Green algae	[34]
*Padina gymnospora*	Brown algae	[62,72]
*Padina tetrastromatica*	Brown algae	[73]
*Porphyra haitanensis*	Red algae	[54,69,70,74]
*Pterocladia capillacea*	Red algae	[75]
*Sargassum* *filipendula*	Brown algae	[76]
*Sargassum henslouianum*	Brown algae	[63]
*Sargassum horneri*	Brown algae	[77]
*Sargassum pallidum*	Brown algae	[47]
*Sargassum tenerrimum*	Brown algae	[78]
*Solieria filiformis*	Red algae	[79]
*Spatoglossum asperum*	Brown algae	[31]
*Turbinaria ornata*	Brown algae	[80]
*Ulva fasciata*	Green algae	[63,81]
*Ulva pertusa*	Green algae	[82]
*Undaria pinnatifida*	Green algae	[83]

**Table 7 polymers-11-01163-t007:** Marine algae source with antiviral activity attributed to SPs.

Marine Algal Source	Algae Group	Reference
*Lithothamnion muelleri*	Red algae	[86]
*Monostroma latisimum*	Green algae	[87]
*Sargassum mcclurei*	Brown algae	[84]
*Sargassum polycystum*	Brown algae	[84]
*Turbinaria ornata*	Brown algae	[84]
*Undaria pinnatifida*	Brown algae	[85]

**Table 8 polymers-11-01163-t008:** Marine algae source with anticancer activity attributed to SPs against cancer cell lines and tumors.

Marine Algal Source	Algae Group	Cancer cell lines/Tumor	Reference
*Acanthophora spicifera*	Red algae	A549	[52]
*Enteromorpha intestinalis*	Green algae	HepG2	[24]
*Fucus evanescens*	Brown algae	RPMI-7951	[91]
*Gayralia oxysperma*	Green algae	U-87 MG	[92]
*Laminaria japonica*	Brown algae	HeLa	[68]
*Laurencia obtusa*	Red algae	THP-1	[71]
*Laurencia papillosa*	Red algae	MCF-7	[93]
*Padina tetrastromatica*	Brown algae	HeLa	[19,88]
*Sargassum horneri*	Brown algae	DLD-1	[90]
*Sargassum plagiophyllum*	Brown algae	HepG2, A549	[89]
*Sargassum wightii*	Brown algae	A549	[25]
*Saccharina cichorioides*	Brown alagae	DLD-1	[91]
*Turbinaria conoides*	Brown algae	MiaPaCa-2, Panc-1	[20]
*Undaria pinnatifida*	Brown algae	DMBA-induced tumor, T-47D	[91,94]

**Table 9 polymers-11-01163-t009:** Marine algae source with gastroprotective activity attributed to SPs.

Marine Algal Source	Algae Group	Reference
*Hypnea musciformis*	Red algae	[14,95]
*Laurencia obtusa*	Red algae	[71]
*Solieria filiformis*	Red algae	[79]

**Table 10 polymers-11-01163-t010:** Marine algae source with immunomodulatory activity attributed to SPs.

Marine Algal Source	Algae Group	Reference
*Ascophyllum nodosum*	Brown algae	[50]
*Chondrus crispus*	Red algae	[98]
*Cladophora glomerata*	Green algae	[99]
*Codium fragile*	Green algae	[100]
*Cystoseira indica*	Brown algae	[95]
*Dictyopteris divaricata*	Brown algae	[59]
*Ecklonia cava*	Brown algae	[97]
*Gracilaria rubra*	Red algae	[65]
*Gracilariopsis lemaneiformis*	Red algae	[66,101]
*Kjellmaniella crassifolia*	Brown algae	[102]
*Nemalion helminthoides*	Red algae	[103]
*Padina tetratromatica*	Brown algae	[104]
*Porphyra haitanensis*	Red algae	[105]
*Sargassum angustifolia*	Brown algae	[96]
*Ulva armoricana*	Green algae	[106]
*Ulva intestinalis*	Green algae	[107]
*Undaria pinnatifida*	Brown algae	[102]

**Table 11 polymers-11-01163-t011:** Other marine algae source and its different pharmacologic activity attributed to SPs.

Marine Algal Source	Algae Group	Pharmacologic Activity	Reference
*Caulerpa prolifera*	Green algae	Osteogenic	[117]
*Cystoseira crinita*	Brown algae	Antihyperlipidemic	[111]
*Enteromorpha prolifera*	Green algae	Antihyperlipidemic	[110]
*Gracilaria cervicornis*	Red algae	Antidiarrheal	[109]
*Hypnea musciformis*	Red algae	Antidiarrheal	[108]
*Laminaria japonica*	Brown algae	Renoprotective, vascular protective	[114,115]
*Nemacystus decipiens*	Brown algae	Antithrombotic	[118]
*Porphyra haitanensis*	Red algae	Anti-cellular senescence	[59]
*Sargassum fusiforme*	Brown algae	Skin protective	[116]
*Sargassum vulgare*	Brown algae	Antithrombotic	[43]
*Turbinaria decurrens*	Brown algae	Hepatoprotective	[112,113]
*Ulva sp.*	Green algae	Skin protective	[119]

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
