# Peer review of "Pharmacologic Application Potentials of Sulfated Polysaccharide from Marine Algae"

_polymers, 2019, doi:10.3390/polym11071163_

Round 1
Reviewer 1 Report
In the present manuscript, the authors intended to review all the potential pharmacological applications of sulfated polysaccharides from marine algae. However, the work developed does not show in depth the data published recently in any of the therapeutic areas mentioned. There is no discussion on the results published by other groups. The authors only enumerate in tables some publications in the field. For example, Antioxidant activity of SPs is the most reported in the literature, but the manuscript only mentions some oxidative stress effects on biomolecules and several chemical assays, without discussión on any specific SP.
In addition to that, the manuscript needs an in-dep English edition revision.
Author Response
Reviewer 1:
1. In the present manuscript, the authors intended to review all the potential pharmacological applications of sulfated polysaccharides from marine algae. However, the work developed does not show in depth the data published recently in any of the therapeutic areas mentioned. There is no discussion on the results published by other groups. The authors only enumerate in tables some publications in the field. For example, Antioxidant activity of SPs is the most reported in the literature, but the manuscript only mentions some oxidative stress effects on biomolecules and several chemical assays, without discussion on any specific SP.
Reply: Thank you for the reviewer’ reminder
In this study, we rewrite some part of therapeutic areas in article.
-line 53 to 54:In a study of Sanjeew et al [4], high molecular weight SPs were isolated from Sargassum horneri and tested both in vitro in RAW 264.7 cells and in vivo model by zebrafish.
-line 58 to 63: LPS-induced zebrafish also showed NO inhibitory activity in the same fraction (IC50=87.12 μg/mL). An earlier research performed on the same brown algae concluded that one of the purified fraction caused a significant decrease in the tumor necrosis factor (TNF-α) secretion and interleukin-10 (IL-10) secretions were seen in LPS-induces RAW 264.7 cell as well [5] as well as NO release and inducible NO synthase (iNOS) were inhibited by the SP fraction of high dose at 200 μg/mL.
-line 70 to 73: Fucoidans from Choonospora minima [7] showed a very similar activity on zebrafish model and RAW 264.7 macrophages cell as it inhibits anti-inflammatory factors such as NO and reactive oxygen species (ROS) production (IC50=27.82±0.88 μg/mL), PGE2 expressions, TNF-α, IL-1β and IL-6.
-line 86 to 90: Fractions extracted from several Phaeophyceae species have exhibited antiangiogenic property in the CAM assay. In the study of Kurup et al [19], the purified sulfated polysaccharides were isolated from Padina tetrastromatica, then named as ESPs-CP (IC50=1.2 mg/mL) has been shown a significant suppression of angiogenesis and a reduction in hemoglobin was also quantified.
-line 96 to 99: This result showed that PSV1 at 50 μg/μL (71.4%) is more effective than SV1 at 100 μg/μL (75.9%), it also supported by the an inhibition of tubulogenesis in rabbit aorta endothelial cells [21].
-line 117 to 118: AgNP has been showed that IC50=10 μg/mL in Balb/3T3 mouse embryo fibroblasts and antimicrobial activity.
-line 123 to 124:The dose of ulvan at 20 μg/mL of S. aureus has been showed that bacterial growth inhibition (9%) of treated with lysozyme (200 μg/mL)[27].
-line 149 to 152: The SVSP administration indicated a significant decreasing in concentrations of serum for T-Ch, TG, LDL-Ch (24.91%, 46.78% and 51.75%, respectively) in the animal model of diabetic. This is the significant increasing related in the serum HDL-Chconcentration (53.12%) which compared to untreated diabetics [44].
-line 160 to 161: An administration of fucoidan at 200 and 1200 mg/kg body weight/day could significantly reduce the blood glucose level by 22% and 34%, respectively.
-line 168 to 170: S. pallidum at the high concentration of 1 mg/mL, the highest inhibitory rates of SPP, S-SPP1–4, S-SPP1–6 and S-SPP1–8 were 76.3%, 96.9%, 97.6% and 98.4%, respectively. The inhibition of sulfated derivatives has been showed much higher than native SPP.
-line 179 to 192: It is well known that free radicals which including oxygen free radicals and non-radical derivatives of oxygen. They are the side products of normal metabolism. Superoxide and OH-radicals are the important reactive oxygen species in the body. One of the main causes of cytotoxicity is superoxide free radicals, because it is the first oxygen free radical produced in the body and lasts longer than other free radicals. Hydroxyl groups are the most active free radicals that attack all biomolecules by initiating a free radical chain reaction. However, excess free radicals are potentially harmful to various biomolecules through lipid peroxidation, DNA damage and inhibition of protein synthesis. This damage accelerates aging and causes many diseases such as cancer and tumor [99]. Many genera of algae have been reported that their antioxidant activity. The algae antioxidants which including phycobiliproteins, phlorotannins, carotenoids, sulfated polysaccharides, scytonemin and mycosporin-like amino acids also have been shown in the literatures by their biologically significant activities. Fucoidan, porphyran, carrageenans, and ulvan (Figure 1) are part of SPs which have been found for their antioxidant activity such as DPPH, ABTS, NO, super oxide and hydroxyl radicals.
-line 200 to 201:Only natural products (compounds, extracts) have antioxidant properties.
-line 217 to 221: The IC50 range is 27-170 and 90-120 mg/mL in the isolates and challenge virus, respectively. The subpopulation presence of drug-resistant viruses was confirmed the lungs and bronchoalveolar lavage fluids of immunocompromised mice administered with oseltamivir, the IC50 range is 0.16-42 and 0.18-0.30 mg/mL in the isolates and challenge virus, respectively [85].
-line 236 to 240: CFs concentration of and their fractions, it found that the cells could be dead. An exhibition of F3 showed the less cytotoxicity than F1 and F2 with around 30% inhibition at the concentration of 2 mg/ml. The literature also indicated that F1 and F2 of the extract fucoidan whicj isolated from Laminaria japonica has been found the less activity than the fucoidans from Undaria pinnatifida with the IC50=0.4 mg/mL [61].
-line 244 to 246: The ESPs-CP decreased the HeLa cells viability in a dose-dependent manner and at 1.2 mg/ml concentration that showed 50% reduction (IC50) in the viability [88,19].
-line 259 to 262: MgONPs has been showed that cytotoxicity against A549 cell line in a dose dependent manner with the IC50=37.5 ± 0.34 μg/mL [25]. Red algae Acanthopora specifica purified fraction has been tested in the cell line and the exhibition of cytotoxicity at the range of concentrations at 100 to 1000 μg/mL [57]. -line 265 to 266: The F2 fraction has been showed that more effective anticancer activity in both of HepG2 and A549 cells with IC50 values of 600 μg/mL and 700 μg/mL, respectively [89].
-line 271 to 272: The fractions of HE1, HE4 and HE7 were obtained from polysaccharide of EHEM showed the cytotoxicity against HepG2 with IC50=73.1, 42.6 and 76.2 μg/mL, respectively.
-line 281 to 285: The polysaccharide fractions of SHP30 and SHP80 showed that some inhibition effect at the concentrations from 0.5 to 6.0 mg/mL. The SHP30 fraction contented highest sulfate and intermediate molecular weight which showed relatively higher inhibition of 51.92% at the concentration of 2mg/mL than SHP80 at all concentrations. Otherwise, the SHP60 fraction conduced to MKN45 cells to grow at the concentrations ranges of 0.5 to 6 mg/mL [90].
-line 298 to 300: The gastroprotective effect (p < 0.01) of SPs has been also observed with the gastric ulcer inhibition of 63.44%, 78.42% and 82.15% at the concentration ranges of 25, 50 and 100 mg/kg, respectively [59].
-line 306 to 307: In vivo model was treated with sulfated-polysaccharide (PLS) fraction of H. musciformis at the doses of 3, 10, 30, and 90 mg/kg.
-line 328 to 331: All crude and fractionated polysaccharides of C. indica were tested on RAW 264.7 cells at the concentration ranges of 10 to 50 μg/mL, CIF2 polysaccharide is a proliferation stimuli on RAW 264.7 cells, it can be enhanced cell growth around 25% [96].
-line 333 to 334: The DDSP as a proliferation stimuli on RAW 264.7 cells at the concentration ranges of 50 to 400 μg/mL [55].
-line 341 to 343: The crude and fractions of S. angustifolium effect on the proliferation of RAW 264.7 cells at concentrations ranges of 10 to 50 μg/mL. Fraction F2 is a proliferation stimuli on RAW 264.7 cells at concentration of 50 μg/mL (p<0.05) [98].
-line 351 to 352: The phosphory-lated ERK expression was no showed any significant differences at the ascophyllan concentration ranges of 0 to1000 μg/mL [99].
-line 354 to 355: And literature was indicated that SP led to the lymphocytes proliferation at the concentration ranges of 25, 50, and 150 μg/mL [100].
-line 360 to 362: Efficacy of SPs from Hypnea musciformis (PLS 90 mg/Kg) [110] and Gracilaria cervicornis (PLS of enzymatic extraction 25 to 30 g) [111] as antidiarrheal were tested in rats and mice respectively.
-line 370 to 372: Recent studies also suggest that Enteromorpha prolifera (The administration of EP 200 mg/kg body weight/daily) [112] and Cystoseira crinita (The administration of CCSP 200 mg/kg of body weight/daily) [113] both possess antihyperlipidemic on rats.
-line 395 to 396: The SV1 of S. vulgare was evaluated inflammatory activity at the concentration ranges of 10, 30 and 50 mg/kg using carrageenan-induced acute rat model [43]

Reviewer 2 Report
The article provides a comprehensive review of the experimental work performed on sulfated polysaccharides from algae and highlights a large number of algae species investigated with regard to their bioactivity. Some important matters have however to be addressed before publication:
1. A presentation of the chemical structure of sulfated polysaccharides from algae, as well as a classification from a structural point of view should be included at the beginning of the review. In their paper, the authors discuss the bioactivity of fucoidan, ulvan, rhamnan-type sulfated polysaccharides, structural information in this regard is necessary in order to improve comprehension by the readership.
2. An extensive English grammar correction should be performed, as many formulation errors occur throughout the text. Some examples are mentioned below:
- line 54: “Sargassum horneri containing high molecular weight SPs were isolated...” – Sargassum horneri is an algae species and cannot be isolated – please rephrase to: high molecular weight SPs were isolated (or extracted) from Sargassum horneri and tested...
-line 59: “On the same brown algae on an earlier study, one of the purified fraction caused....” could be rephrased to “An earlier research performed on the same brown algae concluded that one of the purified fraction caused.....”
- line 87: “Padina tetrastromatica ethanolic SP-column purified have shown a significant suppression....” it is unclear what has actually been tested
- lines 173-174: “Across all families of marine algae, antioxidant species are present” please rephrase as only natural products (compounds, extracts) have antioxidant properties, not species themselves.
3. A confusion between algae groups and families is present throughout the text:
Phaeophyceae in line 85 is not a botanical family, but comprises all brown Algae; thus it is advisable to formulate: “Fractions extracted from several Phaeophyceae species have exhibited...”
The same confusion is present in all tables, where red algae and brown algae are considered “families”. The reformulation to “Algae group” should be performed.
4. Quantitative data regarding bioactivity lacks from the review. Although the type of bioactivity and algae species are thoroughly reviewed, the therapeutic potential of any natural product is highly dependent on the concentration/dosage needed to elicit a pharmacological effect. Some examples and discussions in this regard should be inserted.
Author Response
Reviewer 2
1. A presentation of the chemical structure of sulfated polysaccharides from algae, as well as a classification from a structural point of view should be included at the beginning of the review. In their paper, the authors discuss the bioactivity of fucoidan, ulvan, rhamnan-type sulfated polysaccharides, structural information in this regard is necessary in order to improve comprehension by the readership.
Reply:
Thank you for reviewer reminder. We added Figure 1 at Line 205.
2. An extensive English grammar correction should be performed, as many formulation errors occur throughout the text. Some examples are mentioned below:
- line 54: “Sargassum horneri containing high molecular weight SPs were isolated...” – Sargassum horneri is an algae species and cannot be isolated – please rephrase to: high molecular weight SPs were isolated (or extracted) from Sargassum horneri and tested...
Reply:
The sentence has been rewritten as ‘’high molecular weight SPs were isolated from Sargassum horneri and tested both in vitro in RAW 264.7 cells and in vivo model by zebrafish.’’
-line 59: “On the same brown algae on an earlier study, one of the purified fraction caused....” could be rephrased to “An earlier research performed on the same brown algae concluded that one of the purified fraction caused.....”
Reply:
The sentence has been rewritten as An earlier research performed on the same brown algae concluded that one of the purified fraction caused a significant decrease in the tumor necrosis factor (TNF-α) secretion and interleukin-10 (IL-10) secretions were seen in LPS-induces RAW 264.7 cell as well [5] as well as NO release and inducible NO synthase (iNOS) were inhibited by the SP fraction.
- line 87: “Padina tetrastromatica ethanolic SP-column purified have shown a significant suppression....” it is unclear what has actually been tested
Reply:
The sentence has been rewritten as ‘’the purified sulfated polysaccharides from Padina tetrastromatica were then named as Ethanolic Sulfated Poly-saccharides-Column Purified (ESPs-CP) have been shown a significant suppression of angiogenesis and a reduction in hemoglobin was also quantified. ‘’
- lines 173-174: “Across all families of marine algae, antioxidant species are present” please rephrase as only natural products (compounds, extracts) have antioxidant properties, not species themselves.
Reply:
The sentence has been rewritten as Only natural products (compounds, extracts) have antioxidant properties at Line 200.
3. A confusion between algae groups and families is present throughout the text:
Phaeophyceae in line 85 is not a botanical family, but comprises all brown Algae; thus it is advisable to formulate: “Fractions extracted from several Phaeophyceae species have exhibited...”
The same confusion is present in all tables, where red algae and brown algae are considered “families”. The reformulation to “Algae group” should be performed.
Reply:
The sentence has been rewritten as Fractions extracted from several Phaeophyceae species have exhibited antiangiogenic property in the CAM assay. In the study of Kurup et al [19], the purified sulfated polysaccharides were isolated from Padina tetrastromatica, then named as ESPs-CP (IC50=1.2 mg/mL) has been shown a significant suppression of angiogenesis and a reduction in hemoglobin was also quantified. Line 86-90.
4. Quantitative data regarding bioactivity lacks from the review. Although the type of bioactivity and algae species are thoroughly reviewed, the therapeutic potential of any natural product is highly dependent on the concentration/dosage needed to elicit a pharmacological effect. Some examples and discussions in this regard should be inserted.
Reply:
Thank you for the reviewer’ reminder, we revised some part of therapeutic areas the article.
-line 53 to 54:In a study of Sanjeew et al [4], high molecular weight SPs were isolated from Sargassum horneri and tested both in vitro in RAW 264.7 cells and in vivo model by zebrafish.
-line 58 to 63: LPS-induced zebrafish also showed NO inhibitory activity in the same fraction (IC50=87.12 μg/mL). An earlier research performed on the same brown algae concluded that one of the purified fraction caused a significant decrease in the tumor necrosis factor (TNF-α) secretion and interleukin-10 (IL-10) secretions were seen in LPS-induces RAW 264.7 cell as well [5] as well as NO release and inducible NO synthase (iNOS) were inhibited by the SP fraction of high dose at 200 μg/mL.
-line 70 to 73: Fucoidans from Choonospora minima [7] showed a very similar activity on zebrafish model and RAW 264.7 macrophages cell as it inhibits anti-inflammatory factors such as NO and reactive oxygen species (ROS) production (IC50=27.82±0.88 μg/mL), PGE2 expressions, TNF-α, IL-1β and IL-6.
-line 86 to 90: Fractions extracted from several Phaeophyceae species have exhibited antiangiogenic property in the CAM assay. In the study of Kurup et al [19], the purified sulfated polysaccharides were isolated from Padina tetrastromatica, then named as ESPs-CP (IC50=1.2 mg/mL) has been shown a significant suppression of angiogenesis and a reduction in hemoglobin was also quantified.
-line 96 to 99: This result showed that PSV1 at 50 μg/μL (71.4%) is more effective than SV1 at 100 μg/μL (75.9%), it also supported by the an inhibition of tubulogenesis in rabbit aorta endothelial cells [21].
-line 117 to 118: AgNP has been showed that IC50=10 μg/mL in Balb/3T3 mouse embryo fibroblasts and antimicrobial activity.
-line 123 to 124:The dose of ulvan at 20 μg/mL of S. aureus has been showed that bacterial growth inhibition (9%) of treated with lysozyme (200 μg/mL)[27].
-line 149 to 152: The SVSP administration indicated a significant decreasing in concentrations of serum for T-Ch, TG, LDL-Ch (24.91%, 46.78% and 51.75%, respectively) in the animal model of diabetic. This is the significant increasing related in the serum HDL-Chconcentration (53.12%) which compared to untreated diabetics [44].
-line 160 to 161: An administration of fucoidan at 200 and 1200 mg/kg body weight/day could significantly reduce the blood glucose level by 22% and 34%, respectively.
-line 168 to 170: S. pallidum at the high concentration of 1 mg/mL, the highest inhibitory rates of SPP, S-SPP1–4, S-SPP1–6 and S-SPP1–8 were 76.3%, 96.9%, 97.6% and 98.4%, respectively. The inhibition of sulfated derivatives has been showed much higher than native SPP.
-line 179 to 192: It is well known that free radicals which including oxygen free radicals and non-radical derivatives of oxygen. They are the side products of normal metabolism. Superoxide and OH-radicals are the important reactive oxygen species in the body. One of the main causes of cytotoxicity is superoxide free radicals, because it is the first oxygen free radical produced in the body and lasts longer than other free radicals. Hydroxyl groups are the most active free radicals that attack all biomolecules by initiating a free radical chain reaction. However, excess free radicals are potentially harmful to various biomolecules through lipid peroxidation, DNA damage and inhibition of protein synthesis. This damage accelerates aging and causes many diseases such as cancer and tumor [99]. Many genera of algae have been reported that their antioxidant activity. The algae antioxidants which including phycobiliproteins, phlorotannins, carotenoids, sulfated polysaccharides, scytonemin and mycosporin-like amino acids also have been shown in the literatures by their biologically significant activities. Fucoidan, porphyran, carrageenans, and ulvan (Figure 1) are part of SPs which have been found for their antioxidant activity such as DPPH, ABTS, NO, super oxide and hydroxyl radicals.
-line 200 to 201:Only natural products (compounds, extracts) have antioxidant properties.
-line 217 to 221: The IC50 range is 27-170 and 90-120 mg/mL in the isolates and challenge virus, respectively. The subpopulation presence of drug-resistant viruses was confirmed the lungs and bronchoalveolar lavage fluids of immunocompromised mice administered with oseltamivir, the IC50 range is 0.16-42 and 0.18-0.30 mg/mL in the isolates and challenge virus, respectively [85].
-line 236 to 240: CFs concentration of and their fractions, it found that the cells could be dead. An exhibition of F3 showed the less cytotoxicity than F1 and F2 with around 30% inhibition at the concentration of 2 mg/ml. The literature also indicated that F1 and F2 of the extract fucoidan whicj isolated from Laminaria japonica has been found the less activity than the fucoidans from Undaria pinnatifida with the IC50=0.4 mg/mL [61].
-line 244 to 246: The ESPs-CP decreased the HeLa cells viability in a dose-dependent manner and at 1.2 mg/ml concentration that showed 50% reduction (IC50) in the viability [88,19].
-line 259 to 262: MgONPs has been showed that cytotoxicity against A549 cell line in a dose dependent manner with the IC50=37.5 ± 0.34 μg/mL [25]. Red algae Acanthopora specifica purified fraction has been tested in the cell line and the exhibition of cytotoxicity at the range of concentrations at 100 to 1000 μg/mL [57]. -line 265 to 266: The F2 fraction has been showed that more effective anticancer activity in both of HepG2 and A549 cells with IC50 values of 600 μg/mL and 700 μg/mL, respectively [89].
-line 271 to 272: The fractions of HE1, HE4 and HE7 were obtained from polysaccharide of EHEM showed the cytotoxicity against HepG2 with IC50=73.1, 42.6 and 76.2 μg/mL, respectively.
-line 281 to 285: The polysaccharide fractions of SHP30 and SHP80 showed that some inhibition effect at the concentrations from 0.5 to 6.0 mg/mL. The SHP30 fraction contented highest sulfate and intermediate molecular weight which showed relatively higher inhibition of 51.92% at the concentration of 2mg/mL than SHP80 at all concentrations. Otherwise, the SHP60 fraction conduced to MKN45 cells to grow at the concentrations ranges of 0.5 to 6 mg/mL [90].
-line 298 to 300: The gastroprotective effect (p < 0.01) of SPs has been also observed with the gastric ulcer inhibition of 63.44%, 78.42% and 82.15% at the concentration ranges of 25, 50 and 100 mg/kg, respectively [59].
-line 306 to 307: In vivo model was treated with sulfated-polysaccharide (PLS) fraction of H. musciformis at the doses of 3, 10, 30, and 90 mg/kg.
-line 328 to 331: All crude and fractionated polysaccharides of C. indica were tested on RAW 264.7 cells at the concentration ranges of 10 to 50 μg/mL, CIF2 polysaccharide is a proliferation stimuli on RAW 264.7 cells, it can be enhanced cell growth around 25% [96].
-line 333 to 334: The DDSP as a proliferation stimuli on RAW 264.7 cells at the concentration ranges of 50 to 400 μg/mL [55].
-line 341 to 343: The crude and fractions of S. angustifolium effect on the proliferation of RAW 264.7 cells at concentrations ranges of 10 to 50 μg/mL. Fraction F2 is a proliferation stimuli on RAW 264.7 cells at concentration of 50 μg/mL (p<0.05) [98].
-line 351 to 352: The phosphory-lated ERK expression was no showed any significant differences at the ascophyllan concentration ranges of 0 to1000 μg/mL [99].
-line 354 to 355: And literature was indicated that SP led to the lymphocytes proliferation at the concentration ranges of 25, 50, and 150 μg/mL [100].
-line 360 to 362: Efficacy of SPs from Hypnea musciformis (PLS 90 mg/Kg) [110] and Gracilaria cervicornis (PLS of enzymatic extraction 25 to 30 g) [111] as antidiarrheal were tested in rats and mice respectively.
-line 370 to 372: Recent studies also suggest that Enteromorpha prolifera (The administration of EP 200 mg/kg body weight/daily) [112] and Cystoseira crinita (The administration of CCSP 200 mg/kg of body weight/daily) [113] both possess antihyperlipidemic on rats.
-line 395 to 396: The SV1 of S. vulgare was evaluated inflammatory activity at the concentration ranges of 10, 30 and 50 mg/kg using carrageenan-induced acute rat model [43]

Round 2
Reviewer 1 Report
The authors have done a significant improvement of the original manuscript, so it could be accepted in the present form.
Reviewer 2 Report
The modifications performed by the authors improved the quality of the manuscript and its readability. The paper is OK for publication.